# Alkaloid Profiling and Cholinesterase Inhibitory Potential of *Crinum* × *amabile* Donn. (Amaryllidaceae) Collected in Ecuador

**DOI:** 10.3390/plants10122686

**Published:** 2021-12-07

**Authors:** Luciana R. Tallini, Angelo Carrasco, Karen Acosta León, Diego Vinueza, Jaume Bastida, Nora H. Oleas

**Affiliations:** 1Programa de Pós-Graduação em Ciências Farmacêuticas, Faculdade de Farmácia, Universidade Federal do Rio Grande do Sul, Av. Ipiranga 2752, Porto Alegre 90610-000, Brazil; lucianatallini@gmail.com; 2Grup de Productes Naturals, Departament de Biologia, Sanitat i Medi Ambient, Facultat de Farmàcia i Ciències de l’Alimentació, Universitat de Barcelona, Av. Joan XXIII, #27-31, 08028 Barcelona, Spain; jaumebastida@ub.edu; 3Grupo de Investigación de Productos Naturales y Farmacia, Facultad de Ciencias, Escuela Superior Politécnica del Chimborazo, Panamericana Sur km 1 1/2, Riobamba EC060155, Ecuador; acarrasco251@gmail.com (A.C.); kacosta@espoch.edu.ec (K.A.L.); drvinueza@espoch.edu.ec (D.V.); 4Centro de Investigación de la Biodiversidad y Cambio Climático (BioCamb) e Ingeniería en Biodiversidad y Recursos Genéticos, Facultad de Ciencias de Medio Ambiente, Universidad Tecnológica Indoamérica, Machala y Sabanilla, Quito EC170301, Ecuador

**Keywords:** Amaryllidaceae alkaloids, *Crinum* × *amabile*, AChE, BuChE, GC-MS

## Abstract

Natural products are one of the main sources for developing new drugs. The alkaloids obtained from the plant family Amaryllidaceae have interesting structures and biological activities, such as acetylcholinesterase inhibition potential, which is one of the mechanisms used for the palliative treatment of Alzheimer’s disease symptoms. Herein we report the alkaloidal profile of bulbs and leaves extracts of *Crinum × amabile* collected in Ecuador and their in vitro inhibitory activity on acetylcholinesterase (AChE) and butyrylcholinesterase (BuChE) enzymes. Using Gas Chromatography coupled to Mass Spectrometry (GC-MS), we identified 12 Amaryllidaceae alkaloids out of 19 compounds detected in this species. The extracts from bulbs and leaves showed great inhibitory activity against AChE and BuChE, highlighting the potential of Amaryllidaceae family in the search of bioactive molecules.

## 1. Introduction

Natural products are one of the main sources for developing new drugs [1]. The plant family Amaryllidaceae, specifically the Amaryllidoideae subfamily, produces a large and exclusive group of alkaloids that show high structural diversity and biological potential, and are known as Amaryllidaceae alkaloids [2]. This subfamily comprises around 800 species, which are classified into 59 genera, and more than 600 alkaloids have been reported in these plants [3].

*Crinum* L. is the only pantropical genus of Amaryllidaceae and occurs in Africa, America, Asia, and Australia [4]. This genus has been widely applied in folk medicine around the world for similar purposes, which suggests the biological potential of its metabolic profiling [2,5]. Traditionally, *Crinum* plants have been used in the treatment of rheumatism, kidney, bladder, or parasitic infections, aching joints, as expectorants, against inflammation processes, among others [6,7].

Cytotoxic, antimicrobial, antioxidant, and anti-inflammatory activities have been reported for *C. latifolium* [8]. Alkaloids isolated from *C. erubescens* were characterized as antiplasmodial by a nanomolar level of activity [9]. Antiproliferative effects against human tumor cell lines were described for alkaloids obtained from *C. zeylanicum* [10]. Moreover, *C. jagus* ameliorates convulsions and protects mice against kindling development, depression-like behavior, and oxidative stress [11].

*Crinum × amabile* Donn. is an ornamental species that has been traditionally used in Vietnamese folk medicine as an emetic and as a remedy for rheumatism and earache [12]. This plant is used in Ecuador to treat inflammations, especially hemorrhoids. This species has shown in vitro anti-inflammatory activity and cytotoxicity [13].

Alzheimer’s disease (AD) is a progressive degenerative brain disease characterized by loss of memory, reasoning, and learning disabilities, due to a decrease in neuronal activity and a decline in concentration of neurotransmitters in the inter-synaptic space [14,15,16]. The cholinergic hypothesis links memory deficits with a malfunction of the cholinergic system in patients with AD [14]. One of the alternatives to reduce this problem is to increase the concentration of the neurotransmitter acetylcholine (ACh), which can be hydrolyzed by acetylcholinesterase (AChE) and butyrylcholinesterase (BuChE) [16]. In the brain of AD patients, AChE activity tends to decrease while that of BuChE increases, then the importance to development cholinesterase inhibitors that suppress both enzymes [17,18]. The AChE inhibitors such as galanthamine, rivastigmine, tacrine, and physostigmine have a mechanism of action based on the cholinergic hypothesis and partially inhibit AChE [19]. However, it has been reported that these last three molecules have secondary effects on health, such as inducing hepatoxicity and gastrointestinal complications [20].

Galanthamine, probably the most important Amaryllidaceae alkaloid, is FDA approved since 2001 for the palliative treatment of mild to moderate Alzheimer’s disease symptomatology because of its ability as an acetylcholinesterase inhibitor [21,22]. Due to its high cost and low yield, this molecule is still obtained from Amaryllidaceae species such as *Galanthus nivalis*, *Leucojum aestivum*, *Lycoris radiata*, and different species of *Narcissus* by pharmaceutical companies [23]. In recent years, the search for new bioactive structures obtained from Amaryllidadeae species has increased, and the anti-cholinesterase potential of different *Crinum* species has been reported for *C. amabile*, *C. bulbispermum*, *C. erubescens*, *C. jagus*, *C. moorei* and *C. zeylanicum* collected in South America [24,25,26,27].

The aim of this study is to report the first alkaloid profiling of *Crinum × amabile* collected in Ecuador, and its anti-cholinesterase activity. Bulbs and leaves alkaloid extracts of this species were obtained, and then analyzed by GC-MS. The inhibitory potential of both alkaloid extracts against AChE and BuChE was evaluated in vitro, and the results have been discussed.

## 2. Results and Discussion

### 2.1. Alkaloid Profiling

We identified twelve Amaryllidaceae alkaloids in bulbs and leaves extracts of *Crinum × amabile* collected in Ecuador (Table 1). This process was done by comparing their GC-MS spectra and Kovats Retention Index (RI) values with those of authentic Amaryllidaceae alkaloids previously isolated and identified by spectroscopic and spectrometric methods in the Natural Products Laboratory of Barcelona University, Spain. The unidentified structures were evaluated using the NIST 05 Database and comparing them to data from the literature; however, they were unidentifiable.

The majority of the alkaloids identified in the bulbs and leaves of *Crinum × amabile* belong to haemanthamine/crinine- and lycorine-type, and just two belong to galanthamine-type (Table 1 and Figure 1). The haemanthamine/crinine-type alkaloids are enantiomers with a 5,10b-ethano-bridge α- or β-oriented, and circular dichroism (CD) analyses are usually required to confirm the structures [28].

Amaryllidaceae alkaloids are derived from aromatic amino acids L-phenylalanine and L-tyrosine, which contribute to 4′-O-methylnorbelladine metabolization [29]. The ortho–para’ coupling of 4′-O-methylnorbelladine results in the formation of the lycorine-type skeleton, the para–ortho’ phenolic oxidative coupling originates galanthamine-type, whilst para–para’ coupling gives the haemanthamine/crinine-type structures [10]. There are at least 119 lycorine-, 76 haemanthamine-, 85 crinine- and 47 galanthamine-type alkaloids reported in the literature [3].

The single alkaloid content in both samples was quantified and reported as mg GAL·g^−1^ alkaloid extract (AE). The total of alkaloids identified in these samples was 171.8 and 150.0 mg GAL·g^−1^ AE in bulbs and leaves, respectively (Table 1). An amount of 25.7 and 62.6 mg GAL·g^−1^ AE of haemanthamine/crinine-type alkaloids have been quantified in bulbs and leaves, respectively, of this species. High concentrations of lycorine-type alkaloids have been observed in bulbs, while lower concentrations have been described in leaves, 118.9 and 29.8 mg GAL·g^−1^ AE, respectively. Two alkaloids of galanthamine-type group have been identified in bulbs and leaves of *Crinum* × *amabile*, totalizing 15.3 and 11.6 mg GAL·g^−1^ AE, respectively. The alkaloids buphanisine (1), lycorine (10), and 3-*O*-acetylsanguinine (12), respectively, were the most representative alkaloids of these groups identified in these extracts.

Seven compounds detected in *Crinum × amabile* extracts were not identified (see Table 1). A high amount of unidentified structures has been reported in leaves of *C.* × *amabile* collected in Ecuador, totalizing 46.0 mg GAL·g^−1^ AE, while in bulbs it was 11.9 mg GAL·g^−1^ AE. The compound 17, *m/z* 269 [M^+^ = 269] (RI 2685.8) was the most representative of unidentified structures, 12.8 mg GAL·g^−1^ AE. The similarity between the MS spectrum of compound 17 with the fragmentation pattern of oxocrinine [30] suggests it could belong to haemanthamine/crinine-type alkaloid.

In a recent publication, bulbs and leaves alkaloid extracts of *C. amabile* collected in Brazil were fractionated using different chromatographic tools and twenty-five Amaryllidaceae alkaloids were identified in this species by GC-MS and spectroscopic tools, for example, NMR and CD [26]. The authors reported ten alkaloids belonging to haemanthamine/crinine-type in *C. amabile* collected in Brazil, as well as five and four to lycorine- and galanthamine-type, respectively, and four belonging to other Amaryllidaceae alkaloid-groups [26]. In another study, fifteen alkaloids belonging to haemanthamine/crinine-, lycorine- and galanthamine-type have been identified by GC-MS in bulbs and leaves of *C. amabile* collected in Venezuela [27]. The alkaloid profiling of *C. amabile* collected in Ecuador (Table 1) shows high similarity to the one collected in Venezuela, which reported the same alkaloids as Ecuador sample and four other alkaloids: 6-hydroxybuphanidrine, 6-methoxybuphanidrine, crinamine and galanthamine, except for the presence of 3-*O*-acetylsanguinine (12) [27]. Some similarity between the alkaloid profiling of *Phaedranassa dubia* (Amaryllidaceae) also has been reported for samples collected in two different localities of Ecuador [28,31]. Despite that, the same species collected in different places can show differences in their alkaloid profiling, as described in *Rhodophiala andicola* (Amaryllidaceae) collected from two different localities in Chile [32].

Herein, higher yields of alkaloids have been obtained in the leaves extract than that in the bulbs of *Crinum × amabile* collected in Ecuador, 2.52 and 1.94%, respectively (Table 2). Rojas-Vera and co-authors [27] obtained great yields, 4.79%, for alkaloid extract of aerial parts of *C. amabile* collected in Venezuela; however, the authors reported 0.97% for bulbs extract.

### 2.2. AChE and BuChE Inhibitory Activity

The bulbs alkaloid extract of *Crinum × amabile* collected in Ecuador presented better inhibitory activity against AChE than leaves, with IC_50_ values of 1.35 ± 0.13 and 1.67 ± 0.16 μg mL^−1^, respectively (Figure 2). The opposite happened against BuChE, since leaf extract of this species showed higher potential than bulbs extract, with IC_50_ values of 8.50 ± 0.76 and 45.42 ± 3.72 μg mL^−1^, respectively. The presence of galanthamine-type alkaloids in bulbs and leaves of *Crinum × amabile* collected in Ecuador, represented by sanguinine (11) and 3-*O*-acetylsanguinine (12), can be contributing to inhibitory cholinestarese potential of these extracts, as well as the highest amount of buphanisine (1) and lycorine (10) in these extracts (see Table 1).

Molecular docking results of galanthamine and sanguinine show more hydrogen and hydrophobic interactions with the catalytic site of AChE, on 4EY7 and 5HF5 structures, than buphanisine and lycorine [27]. In vitro experiments with the isolated alkaloid sanguinine presented better AChE inhibitory activity than galanthamine [33]. However, in silico assays report that sanguinine exhibits lower energies of reaction in the active sites of AChE and BuChE compared to galanthamine [27,34]. Computational experiments described that sanguinine presented two hydrogen bonds with His447 and Ser203 of AChE (structure 4EY7), which could be responsible for the better in vitro AChE inhibition potential of sanguinine than galanthamine [27]. However, galanthamine presents a better ability to cross the blood-brain barrier than sanguinine [33].

Leaf extract of *C. amabile* collected in Venezuela showed high activity against AChE and BuChE, with IC_50_ values of 0.88 and 4.46 μg mL^−1^, respectively, while bulbs extract presented IC_50_ values of 2.44 μg mL^−1^ against AChE [27]. Tallini and co-authors [26] evaluated the cholinesterase potential of six alkaloids isolated from *C. amabile* collected in Brazil, and none of the tested alkaloids showed BuChE inhibitory activity. Among them, the authors reported that augustine and augustine *N*-oxide had IC_50_ values of 45.26 ± 2.11 and 79.64 ± 5.26 μg mL^−1^, respectively, against AChE, while buphanisine and crinine showed IC_50_ values of 183.31 ± 36.64 and 163.89 ± 15.69 μg mL^−1^, respectively [26].

The anti-cholinesterase activity of the species *C. erubescens* and *C. moorei* collected in Venezuela has been recently reported in literature. Bulbs extract from *C. erubescens* showed a better enzymatic inhibitory effect than *C. moorei* extract, with IC_50_ values of 3.27 and 9.03 μg mL^−1^, against AChE and BuChE, respectively [27]. The AChE inhibitory potential of other species of *Crinum* from Colombia have been published in the year 2015. Extracts of *C. bulbispermum* collected in different localities, Cundinamarca and Santander, from this country showed IC_50_ values of 25.73 ± 1.75 and 107.90 ± 9.98 μg mL^−1^, respectively, while extracts of *C. jagus* and *C. zeylanicum* presented IC_50_ values of 18.28 ± 0.29 and 70.22 ± 0.24 μg mL^−1^, respectively [24,25].

## 3. Materials and Methods

### 3.1. Plant Material

The species Crinum × amabile Donn. was collected during the flowering period in Muisne (Esmeraldas, Ecuador) in 2017. The species was authenticated by Nora H. Oleas, and a specimen voucher (Oleas #1054) has been deposited at the Herbarium HUTI.

### 3.2. Alkaloid Extraction

Bulbs and leaves of *Crinum × amabile* were dried at 40 °C for 7 days. The dried samples were triturated, and the powder was macerated with methanol at room temperature for 3 days, changing the solvent daily (3 × 100 mL), and, trying to increase the extraction potential, it was applied 20 min of ultrasonic baths, 8 times per day. The mash was filtered and evaporated to dryness under reduced pressure. The crude extracts of the bulbs and leaves were acidified with an aqueous solution of sulfuric acid (2%, *v/v*) to pH 2 and, then cleaned with diethyl ether to remove neutral material. The aqueous solution was basified with ammonium hydroxide at 25% (*v/v*) to pH 9–10 and extracted with ethyl acetate to obtain the alkaloid extract (AE).

### 3.3. GC-MS Analysis

Two mg of each sample were dissolved in 1 mL of methanol: chloroform (0.5:0.5, *v/v*) and evaluated using GC-MS (Agilent Technologies 6890N coupled with MSD5975 inert XL; Santa Clara, CA, USA) equipment operating in electron ionization (EI) mode at 70 eV. A Sapiens-X5 MS column (30 m × 0.25 mm i.d., film thickness 0.25 μm; Teknokroma, Barcelona, Spain) was used. One μL of each sample was injected in the equipment using the splitless mode. Codeine (0.05 mg·mL^−1^) was used as the internal standard. The injector and detector temperatures were 250 and 280 °C, respectively, and the flow-rate of carrier gas (He) was 1 mL·min^−1^. The temperature gradient was 12 min at 100 °C, 100–180 °C at 15 °C·min^−1^, 180–300 °C at 5 °C·min^−1^, and 10 min hold at 300 °C.

A calibration curve of galanthamine (10, 20, 40, 60, 80, and 100 μg·mL^−1^) was applied to quantify each single constituent detected in the chromatogram, using codeine (0.05 mg·mL^−1^) as the internal standard. Peak areas were manually obtained, considering selected ions for each compound (usually the base peak of their MS, i.e., *m/z* at 286 for galanthamine and 299 for codeine). The ratio between the values obtained for galanthamine and codeine in each solution was plotted against the corresponding concentration of galanthamine to obtain the calibration curve and its equation (y = 0.0112x − 0.0469; R^2^ = 0.9995). All data were standardized to the area of the internal standard (codeine), and the equation obtained for the calibration curve of galanthamine was used to calculate the amount of each alkaloid. Results are presented as mg GAL (galanthamine), which was finally related to the alkaloid extract (AE). As the peak area does not only depend on the corresponding alkaloid concentration but also on the intensity of the mass spectra fragmentation, the quantification is not absolute.

### 3.4. AChE and BuChE Inhibitory Activity

Cholinesterases inhibitory activities were determined according to Ellman and co-workers [35] with some modifications as by López and co-workers [36]. Stock solutions with 518U of AChE from *Electrophorus electricus* (Merck, Darmstadt, Germany) and BuChE from equine serum (Merck, Darmstadt, Germany), respectively, were prepared and kept at −20 °C. Acetylthiocholine iodide (ATCI), S-butyrylthiocholine iodide (BTCI), and 5,5′-dithiobis (2-nitrobenzoic) acid (DTNB) were obtained from Merck (Darmstadt, Germany). Fifty microliters of AChE or BuChE (both enzymes used at 6.24 U) in phosphate buffer (8 mM K2HPO4, 2.3 mM NaH2PO4, 0.15 NaCl, pH 7.5) and 50 μL of the sample dissolved in the same buffer were added to the wells. The plates were incubated for 30 min at room temperature. Then, 100 μL of the substrate solution (0.1 M Na_2_HPO_4_, 0.5 M DTNB, and 0.6 mM ATCI or 0.24 mM BTCI in Millipore water, pH 7.5) was added. These reagents were obtained from Merck (Darmstadt, Germany). After 10 min, the absorbance was read at 405 nm in a Labsystem microplate reader (Helsinki, Finland). Enzymes activities were calculated as percent compared to a control using a buffer without any inhibitor. Galanthamine served as a positive control. In the first step, the activity of the samples was assessed at 10, 100, and 200 μg·m^−1^ towards both enzymes. Then, the calibration curves of bulbs alkaloid extract (0.05, 0.1, 0.2, 0.5, 1, 5 and 10 μg·mL^−1^) and leaves (0.5, 1, 2, 3, 4, 5, and 7.5 μg·mL^−1^) were applied to obtain IC_50_ values against AChE, while the calibration curves of bulbs alkaloid extract (10, 20, 30, 50, 70, 90, and 110 μg·mL^−1^) and leaves (0.1, 0.5, 1, 5, 10, 25, and 50 μg·mL^−1^) were used to calculate IC_50_ values against BuChE. The cholinesterases inhibitory data were analyzed with the Prism 9 software.

### 3.5. Statistical Analysis

Three independent assays were used to evaluate the cholinesterase activity of *Crinum amabile* alkaloid extracts. Results were analyzed by ANOVA, using the Prism 9 software. Data are expressed as the mean ± standard deviation (SD). Significant results are marked as follows: **** *p* < 0.0001, and ns (not significant). For AChE and BuChE, one-way ANOVA with Dunnett’s multiple comparison test was used to compare the mean of each column with the mean of a control column (galanthamine).

## 4. Conclusions

This is the first report about the alkaloid profile of the species *Crinum × amabile* collected in Ecuador, as well as about the anti-cholinesterase potential of this plant. We identified 12 alkaloids out of 19 detected in these extracts, suggesting the potential of this species as a source of new compounds. The leaves and bulbs extracts collected in Ecuador showed great inhibitory activity against AChE and BuChE, corroborating with the data available in the literature for this same species collected in Venezuela, evidencing the importance of Amaryllidaceae family in the search of bioactive molecules.

## Figures and Tables

**Figure 1 plants-10-02686-f001:**
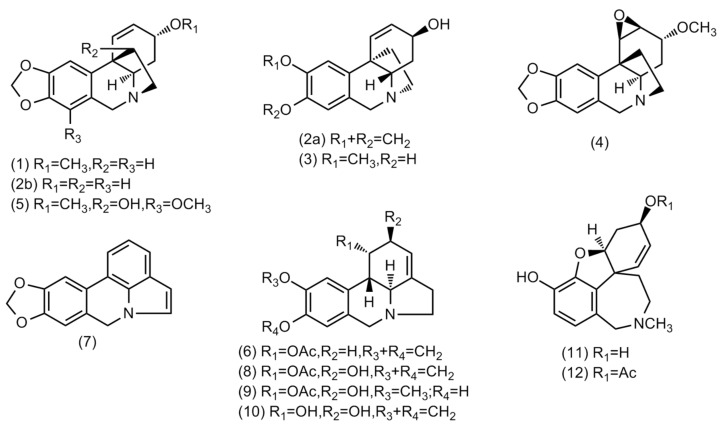
Structures of alkaloids identified in *Crinum × amabile* collected in Ecuador by GC-MS.

**Figure 2 plants-10-02686-f002:**
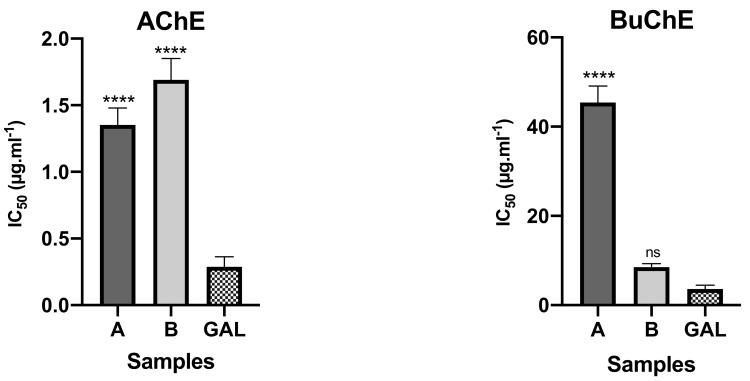
AChE and BuChE inhibitory potential of *Crinum × amabile* collected in Ecuador. A—alkaloid extract of the bulbs; B–alkaloid extract of the leaves; GAL—reference compound; **** *p* < 0.0001; ns—not significant.

**Table 1 plants-10-02686-t001:** GC-MS alkaloid profiling of *Crinum × amabile* collected in Ecuador. Values expressed as mg GAL·g^−1^ AE.

Alkaloid	MS	RI	A	B
**Haemanthamine/Crimine-Type (Total)**			**(25.7)**	**(62.6)**
buphanisine (1) ^a^	285 (100), 270 (34), 254 (35), 215 (84), 115 (31)	2461.2	16.1	23.0
vittatine/crinine (2a/2b) ^a^	271 (100), 228 (25), 199 (63), 187 (60), 115 (30)	2507.3	(-)	6.8
8-*O*-demethylmaritidine (3) ^a^	273 (100), 230 (39), 201 (87), 189 (56), 128 (36)	2542.9	(-)	15.0
augustine (4) ^a^	301 (100), 228 (33), 175 (75), 143 (37), 115 (44)	2600.4	9.6	12.4
ambelline (5) ^a^	331 (97), 299 (52), 287 (100), 260 (90), 241 (51)	2810.9	(-)	5.4
**Lycorine-Type (Total)**			**(118.9)**	**(29.8)**
1-*O*-acetylcaranine (6) ^a^	313 (67), 252 (100), 240 (11), 226 (91), 227 (35)	2554.5	6.4	(-)
11,12-dehydroanhydrolycorine (7) ^a^	249 (65), 248 (100), 190 (25), 163 (7), 123 (5)	2639.8	5.1	5.1
1-*O*-acetyllycorine (8) ^a^	329 (39), 268 (32), 250 (18), 227 (64), 226 (100)	2750.5	(-)	5.9
sternbergine (9) ^a^	331 (43), 270 (29), 252 (12), 229 (76), 228 (100)	2740.0	5.3	(-)
lycorine (10) ^a^	287 (44), 268 (35), 250 (18), 227 (86), 147 (17)	2817.5	102.1	18.8
**Galanthamine-Type (Total)**			**(15.3)**	**(11.6)**
sanguinine (11) ^a^	273 (100), 272 (82), 256 (18), 202 (31), 160 (38)	2453.0	6.4	5.6
3-*O*-acetylsanguinine (12) ^a^	315 (54), 256 (100), 255 (61), 212 (28), 96 (50)	2556.1	8.9	6.0
**Unidentified (Total)**			**(11.9)**	**(46.0)**
UI (ismine-type) ^b^ (13)	269 (-), 268 (2), 240 (8), 226 (22), 225 (100)	2368.5	(-)	6.1
UI (haemanthamine/crinine-type) ^b^ (14)	301 (65), 300 (100), 244 (67), 215 (19), 201 (17)	2497.4	5.8	7.9
UI (tazettine-type) ^b^ (15)	315 (67), 300 (20), 284 (47), 260 (100), 229 (83)	2517.7	(-)	6.8
UI (haemanthamine/crimine-type) ^b^ (16)	301 (66), 286 (8), 270 (10), 246 (100), 231 (43)	2645.0	6.1	(-)
UI (haemanthamine/crimine-type) ^b^ (17)	269 (100), 240 (47), 224 (30), 211 (26), 181 (69)	2685.8	(-)	12.8
UI (galanthamine-type) ^b^ (18)	299 (94), 244 (100) 229 (28), 201 (40), 187 (18)	2758.2	(-)	5.5
UI (haemanthamine/crimine-type) ^b^ (19)	315 (100), 272 (8), 258 (93), 240 (10), 188 (63)	2875.4	(-)	6.9
**Total Alkaloids:**			**171.8**	**150.0**

RI = Kovats Retention Index; A = bulbs; B = leaves; UI = unidentified; ^a^ identified using our homemade Amaryllidaceae alkaloid library; ^b^ proposed structure-type according to the fragmentation pattern; (-) = not reported in the alkaloid profiling.

**Table 2 plants-10-02686-t002:** Yield of alkaloid extract obtained from *Crinum × amabile* collected in Ecuador.

Sample	Dried Plant (g)	Crud Extract (g)	Alkaloid Extract (mg)	Yield (%)
Bulbs	5.00	1.43	27.74	1.94
Leaves	5.26	1.37	34.50	2.52

## Data Availability

Not applicable.

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
