# Peer review of "Alkaloid Profiling and Cholinesterase Inhibitory Potential of Crinum × amabile Donn. (Amaryllidaceae) Collected in Ecuador"

_plants, 2021, doi:10.3390/plants10122686_

Round 1

Reviewer 1 Report

This interesting paper has been well prepared according to authors guideline.  In this paper a new source of cholinesterase inhibitors was described. The obtained results confirm that alkaloids  isolated from Crinnum x amabile Donn. can be successfully applied in the future  in the treatment of Alzheimers disease as natural drug. Alkaloid structures  have been confirmed by using  GC-MS method.  Biological activity were presented and compared as IC50 values. 

There is no reviewer's suggestion. 

Author Response

Thank you for the comments. Even though the reviewer did not suggest any changes, we fixed several minor spells or grammatical issues. 

Reviewer 2 Report

The authors present a profiling study, describing the 12 alkaloids identified in the Crinum x amabile species. 

In the first paragraph of Results and discussion, the author claim that the process was done comparing the current results with a database and literature data, however, no references were added. 

Also, no statistical analysis was described on the MM.

Author Response

All the Amaryllidaceae alkaloids have been identified using our homemade library at Universitat de Barcelona.  This information has been clarified in lines 84-89 and Table 1 (lines 104-105).

Reviewer 3 Report

This manuscript compared the alkaloids profile of the leave and the bulbs of Crinum amabile collected in Ecuador by GC/MS.  Twelve Amaryllidaceae alkaloids out of 19 detected compounds were identified based on the retention index and mass spectrometric data. The quantity of the total alkaloids was measured using galanthamine as a reference. The inhibitory activity of the extract on the acetylcholinesterase (AChE) and butyrylcholinesterase (BuChE) enzymes was also tested. I am not sure if the identification of the 12 alkaloids is confident since no data for structure determination was provided.  The authors should include the GC/MS chromatogram and the MS data in the paper.  The mass spectra may be provided in the supplementary material.  Explanation of the MS fragmentation pathway may help to prove the structure determined. 

The following observations were made

  1. Result and discussion

Table 1. list the major MS data in the table, show the references of MS and Kovats Retention Index (RI) data for the identified alkaloids should be shown. Indicate the methods used to identify each compound, for example, a) match the RI with literature; b) match the MS with literature; c) match the RI and MS of the reference standard.  

Line 122~124 “The compound 17, m/z 269 [M+=269] (RI 2685.8) was the most representative of unidentified structures, 12.8 mg GAL·g−1 AE, and probably belongs to haemanthamine/crinine-type alkaloids due to its fragmentation pattern. “ The MS data for the fragmentation should be provided and explain what patterns suggesting haemanthamine/crinine-type alkaloids.  Literature should be cited to support this claim.

            Line 142 Change “Herein it has been obtained higher yields of alkaloid extract for leaves    than bulbs” to “Herein, higher yields of alkaloids have been obtained in the leaves extract        than that in the bulbs”

            Table 2. “Yield of alkaloid extract obtained from Crinum x amabile collected in Ecuador”, this indicates the yield should be the mass of alkaloids extract over the mass of plant material. However, the yield given is the mass of alkaloids extract over the mass of crude extract, which is not very meaningful since it depends on the extract method.

Line 150: “The bulbs extract of Crinum x amabile collected in Ecuador presented better inhibitory 150 activity against AChE than leaves extract, with IC50 values of 1.35 ± 0.13 and 1.67 ± 0.16 151 μg.ml-1, respectively”, is the bulbs extract the crude extract or the alkaloid extract?

  1. Materials and Methods

Line 198 – 199: “applying 20 minutes of ultrasonic baths, 8 times per day.”  Eight times of ultrasonic is an unusual extract method.  Please explain why this is used.

Line 237: “50 μl of the sample dissolved in 237 the same buffer.. ”. Is the sample directly dissolved in the buffer? Is DMSO utilized to assist the soluble of the sample?  What’s the concentration of the sample?

Author Response

We thank the suggestions made by the reviewer. Besides them, we also fixed some grammar and spell issues.

Here we included the comments to the reviewer´s suggestions:

Result and Discussion

Comment: Table 1. list the major MS data in the table, show the references of MS and Kovats Retention Index (RI) data for the identified alkaloids should be shown. Indicate the methods used to identify each compound, for example, a) match the RI with literature; b) match the MS with literature; c) match the RI and MS of the reference standard.

Answer: The major MS data have been included in Table 1 (column MS). The methods used to identify each compound are explained in lines 84-89. The sentence “…Identified using our homemade Amaryllidaceae alkaloid library…” has been included in Table 1 (line 104-105).

Comment: Line 122~124 “The compound 17, m/z 269 [M+=269] (RI 2685.8) was the most representative of unidentified structures, 12.8 mg GAL·g−1 AE, and probably belongs to haemanthamine/crinine-type alkaloids due to its fragmentation pattern. “ The MS data for the fragmentation should be provided and explain what patterns suggesting haemanthamine/crinine-type alkaloids.  Literature should be cited to support this claim.

Answer: The major MS data of compound 17 have been included in Table 1 and the sentence “…the similarity between the MS spectrum of compound 17 with the fragmentation pattern of oxocrinine [Ali et al., 1986] suggests it could belongs to haemanthamine/crinine-type alkaloid…” has been added in lines 126-128.

Comment: Line 142 Change “Herein it has been obtained higher yields of alkaloid extract for leaves than bulbs” to “Herein, higher yields of alkaloids have been obtained in the leaves extract than that in the bulbs”

Answer: fixed as suggested.

Comment: Table 2. “Yield of alkaloid extract obtained from Crinum x amabile collected in Ecuador”, this indicates the yield should be the mass of alkaloids extract over the mass of plant material. However, the yield given is the mass of alkaloids extract over the mass of crude extract, which is not very meaningful since it depends on the extract method.

Answer: We chose to give the yield according to mass alkaloid extract because it is comparable to the available literature. The readers can extract different information using the data shown in Table 2.

Comment: Line 150: “The bulbs extract of Crinum x amabile collected in Ecuador presented better inhibitory activity against AChE than leaves extract, with IC50 values of 1.35 ± 0.13 and 1.67 ± 0.16 151 μg.ml-1, respectively”, is the bulbs extract the crude extract or the alkaloid extract?

Answer: fixed as suggested.

Materials and Methods

Comment: Line 198 – 199: “applying 20 minutes of ultrasonic baths, 8 times per day.”  Eight times of ultrasonic is an unusual extract method.  Please explain why this is used.

Answer: The sentence “… trying to increase the extraction potential, it was applied 20 minutes of ultrasonic baths…” was included (line: 204-205).

Comment: Line 237: “50 μl of the sample dissolved in the same buffer.. ”. Is the sample directly dissolved in the buffer? Is DMSO utilized to assist the soluble of the sample?  What’s the concentration of the sample?

Answer: Yes, the sample was directly dissolved in the buffer. DMSO was not used. More information about the concentration of the samples has been included in lines 251-256.

Reviewer 4 Report

The work seems to have been carried out competently and properly. The article is clear, well organized and the statistical analysis well done.

Author Response

The reviewer did not suggest any changes.

Round 2

Reviewer 3 Report

A version highlighting the changes should be provided.